# Ti64/20Ag Porous Composites Fabricated by Powder Metallurgy for Biomedical Applications

**DOI:** 10.3390/ma15175956

**Published:** 2022-08-29

**Authors:** Luis Olmos, Ana S. Gonzaléz-Pedraza, Héctor J. Vergara-Hernández, Jorge Chávez, Omar Jimenez, Elena Mihalcea, Dante Arteaga, José J. Ruiz-Mondragón

**Affiliations:** 1INICIT, Universidad Michoacana de San Nicolás de Hidalgo, Fco. J. Mujica S/N, Morelia C.P. 58060, Mexico; 2División de Estudios de Posgrado e Investigación, Tecnológico Nacional de México/I.T. Morelia, Av. Tecnológico #1500, Colonia Lomas de Santiaguito, Morelia C.P. 58120, Mexico; 3Departamento de Ingeniería Mecánica Eléctrica, CUCEI, Universidad de Guadalajara, Blvd. Marcelino García Barragán # 1421, Guadalajara C.P. 44430, México; 4Departamento de Ingeniería de Proyectos, Universidad de Guadalajara, José Guadalupe Zuno # 48, Los Belenes, Zapopan C.P. 45100, Mexico; 5Centro de Geociencias, Universidad Nacional Autónoma de México, Blvd. Juriquilla No. 3001, Querétaro C.P. 76230, Mexico; 6Corporación Mexicana de Investigación en Materiales SA de CV, Calle Ciencia y Tecnología 790, Fracc. Saltillo 400, Saltillo C.P. 25290, Mexico

**Keywords:** porous composites, liquid phase sintering, biomedical materials, permeability, computed microtomography

## Abstract

We present a novel Ti64/20Ag highly porous composite fabricated by powder metallurgy for biomedical applications and provide an insight into its microstructure and mechanical proprieties. In this work, the Ti64/20Ag highly porous composites were successfully fabricated by the space holder technique and consolidated by liquid phase sintering, at lower temperatures than the ones used for Ti64 materials. The sintering densification was evaluated by dilatometry tests and the microstructural characterization and porosity features were determined by scanning electron microscopy and computed microtomography. Permeability was estimated by numerical simulations on the 3D real microstructure. Mechanical properties were evaluated by simple compression tests. Densification was achieved by interparticle pore filling with liquid Ag that does not drain to the large pores, with additional densification due to the macroscopical deformation of large pores. Pore characteristics are closely linked to the pore formers and the permeability was highly increased by increasing the pore volume fraction, mainly because the connectivity was improved. As expected, with the increase in porosity, the mechanical properties decreased. These results permitted us to gain a greater understanding of the microstructure and to confirm that we developed a promising Ti64/20Ag composite, showing E of 7.4 GPa, σ_y_ of 123 MPa and permeability of 3.93 × 10^−11^ m^2^. Enhanced adaptability and antibacterial proprieties due to Ag were obtained for bone implant applications.

## 1. Introduction

Ti6Al4V (Ti64) alloy is one of the most used materials to fabricate and design devices and bone implants for medical applications. This is because of its excellent mechanical properties, low weight, biocompatibility and good corrosion resistance [1,2,3]. Nonetheless, Ti64 alloy has two main disadvantages for bone implant applications: high stiffness (110 GPa) compared to the human bones (7–30 GPa) and a low antibacterial activity, which means that post-operative infections can appear by the development of bacterial colonies, generating serious complications that can lead to failure of the implants. In order to reduce the stiffness of the Ti64 alloy, porous materials called “scaffolds” have been widely studied by different techniques such as additive manufacturing (AM) [4,5], partial powder sintering [6,7], and the space holder technique [8,9,10]. Although AM is the most outstanding, it is still more expensive with respect to the space holder technique. In addition, fabrication of composites by AM is more complicated than conventional sintering. Thus, highly porous Ti64 has been obtained by the space holder reaching stiffness values similar to human bones [8,9,10]. Furthermore, the Young’s modulus can be roughly predicted as a function of the pore formers used by following a power law, as shown by Cabezas et al. [8]. Moreover, the porosity is also needed to improve the osseointegration because a connected porosity allowing corporal fluid through it with the nutrients enhances the formation of new bone cells, accelerating the bone ingrowth [11,12]. The permeability values measured for human bones were experimentally measured, and a wide range, 10^−8^ to 3.10^−11^ m^2^, was reported, which depend on the kind of bone, as reported by Nauman et al. [13]. Although permeability can be improved by increasing the pore size, as shown in scaffolds fabricated by AM [14,15], the pore size also plays a role in the bone ingrowth. Nevertheless, the optimal pore size is a controversial issue in the literature. While some authors have shown that new bone cells can be developed in samples with pores smaller than 10 µm [16], others have demonstrated that bone can grow in components fabricated by AM with pores larger than 900 µm [17]. Therefore, the best combination of pore size and permeability can be tailored by the space holder technique with a combination of different parameters during sintering.

The poor antibacterial activity of the Ti64 can lead to bone damage by circumferential bone loss that may also compromise the joint with the implant. Consequently, the bone’s mechanical strength is reduced, which leads to the failure of the implant [18,19]. With the aim of overcoming this flaw, elements such as silver (Ag) and copper (Cu), that have demonstrated their excellent antibacterial properties, were added, mainly to the pure Ti [20,21,22,23,24,25]. It was pointed out that adding 5 wt.% of Ag to the Ti significantly reduced the bacterial activity [23]. Ag presents antibacterial activity against a broad-spectrum of microbes including antibiotic-resistant bacterial strains, is stable in physiological medium, and is nontoxic for human cells up to a critical concentration. The antibacterial action of this noble metal is a complex mechanism; various theories have been proposed in the literature. There are two different theories concerning the mechanisms of antibacterial action of the Ag metal. The most widely accepted is that Ag metal ions are released in the biological fluids and kill the bacteria. These Ag ions react with the oxygen to form reactive oxygen species that might affect the DNA, as they oxidize and modify certain cellular components and inhibit them from accomplishing their original functions, as well as the cell membranes and membrane proteins. The dissolution of Ag ions could also generate free radicals on the titanium surface that destroy the bacterial structure and, in this manner, generate the so-called silver ion sterilization [25,26]. The second mechanism is the contact sterilization; the presence of silver on the surface of the titanium generates normal physiological metabolic disorders of the bacterial cell membrane, therefore leading to bacteria death [27]. Shi et al. [28] showed that the Ti_2_Ag phase found on the surface of Ti–Ag alloys, in a contact sterilization mode, plays a larger role in the antibacterial ability than the one generated by the released silver ions.

Ti–Ag alloys are mainly fabricated by casting; nevertheless, it is not easy to achieve homogeneous alloys because of the high difference in melting points of Ti (1668 °C) and Ag (961 °C). Therefore, the powder metallurgy technique offers an alternative to fabricating such alloys by reducing the processing temperatures [23,27,29,30]. This is possible since titanium is immiscible with various metals [31], e.g., Ag [32]. Due to the above-mentioned process, a liquid phase is formed, which drives the densification process during sintering. It was reported that the optimal concentration of Ag in Ti is between 20 and 25 wt.% [33] and Ag in Ti64 alloy is 20 wt.% [34]. This leads to almost fully dense materials and to stable TiAg and Ti_2_Ag phases [23,32]. Nonetheless, the elastic modulus of such materials increased (110–130 GPa) [35], which is a great disadvantage for the materials for bone implant applications.

Therefore, the objective of this paper is to develop highly porous Ti64/20Ag composites that can combine antibacterial activity with low mechanical strength, suitable for bone implants. To achieve our goal, porous composites are fabricated using the space holder technique and liquid state sintering. Particular attention was paid to the effect produced by the large pores on the sintering kinetics, microstructure, permeability, and mechanical properties of the sintered materials.

## 2. Materials and Methods

### 2.1. Sample Preparation

To prepare the Ti6420Ag porous composites, Ti64, Ag and ammonium bicarbonate powders were used. The first ones were spherical with a particle size distribution lower than 45 µm. These powders were furnished by Raymor Company, located in Quebec, Canada. The Ag powders had an irregular shape with a particle size distribution between 1 and 5 µm, and they were furnished by the Sigma Aldrich company. These two powders were used to obtain a composite of Ti64 with 20 vol.% of Ag by mixing them in a TURBULA^®^ shaker mixer during 30 min under dry conditions. The ammonium bicarbonate powders had an irregular shape, and they were sieved to have a particle size distribution between 100 and 200 µm. These powders were furnished by Alfa Aesar and used as pore formers, thus, they were mixed with the Ti6420Ag mixture of powders for 30 min. The quantity of pore formers was varied from 30 to 50 vol.% with respect to the solid of the Ti6420Ag composite. After that, 1 wt.% of PVA was added to the mixture of the three powders as a binder to bring a higher mechanical strength to the green samples. Then, the mixture of powders was poured into an 8 mm cylindrical stainless steel die to uniaxially press the mixture of powders at 450 MPa. This was carried out on an 1150 Instron Universal machine. In order to obtain the porous composites, the pore formers were thermally evaporated by heating the samples at 5 °C/min up to 180 °C at a holding time of 6 h under high purity argon. Next, the samples were introduced into a vertical dilatometer Linseis L75, and the PVA was eliminated by heating the samples at 10 °C/min up to 500 °C with a plateau of 45 min. Finally, the samples were sintered at 1100 °C with a heating rate of 10 °C/min and a soaking time of 5 min in the vertical dilatometer. The sintering thermal cycle was determined in a previous work [34].

The density of the sintered samples was estimated by weighting and measuring their volume and the theoretical density of the Ti6420Ag composite was calculated by the rule of mixtures by using the theoretical density of Ti64 (4.45 g/cm^3^) and Ag (10.49 g/cm^3^), which led to a value of 5.658 g/cm^3^ for the composite with 20 vol.% of Ag.

### 2.2. Microstructural Characterization

The microstructure of the sintered porous composites was observed by scanning electron microscopy (SEM). For that, the samples were half-cut and the transversal section was polished with SiC papers, whereas 1 µm alumina powder was used to obtain a mirror-like surface. Surface analysis was performed using a Tescan, Mira 3 LMU Field Emission Scanning Electron Microscope (FE-SEM). As the SEM images bring only 2D information, 3D images with a voxel size resolution of 1 µm were acquired with the aim of determining the Ag distribution in 3D. To achieve such voxel resolution, samples were cut in prisms of 0.7 mm × 0.7 mm × 4 mm. The 3D image acquisition was performed using a Zeiss Xradia 510 Versa 3D X-ray microscope with a beam intensity of 140 kV, which was high enough to pass through the Ti64/20Ag porous samples of around 0.7 mm thick. An X-ray detector of 20× was used to collect 1600 projections with a CCD camera of 1024 × 1024 pixels by rotating the sample a total of 360°. In addition, 3D images of the whole sample with a 6.5 µm resolution were also acquired to determine the porosity features, such as pore size distribution, connectivity, pore volume, etc. For such images, the beam intensity used was 160 kV with an X-ray detector of 0.4x, used to collect 1600 projections all around the sample.

### 2.3. Permeability Evaluation

The permeability of porous samples was estimated by numerical simulations on the real microstructure issued from the 3D images, acquired with a voxel size of 6.5 µm, and in this way, we obtained the whole image of the porosity. For that, the module of “Absolute Permeability Experiment Simulation” in the Avizo^®^ software was used. This module works with binary images containing the pores with an intensity of 1 and the solid with the intensity of 0. Simulations were carried out by solving the Navier–Stokes equations with an iterative method that is stopped with an error criterion of 10^−5^. The simulations take into account the Darcy law, a Newtonian fluid with a steady state laminar flow where it is possible to introduce the fluid viscosity and the boundary conditions of inlet and outlet pressure. In this work, the inlet and outlet pressures were 130 and 100 kPa, respectively, and the fluid viscosity was 0.045 Pa, which mimics that of human blood. As the simulation module demands a lot of computational requirements, it was necessary to determine a minimum representative volume (MRV) to reduce the computational time of the permeability simulations. For that, the methodology proposed by Okuma et al. [36] was followed, finding a MRV of 250^3^ voxels^3^. Therefore, the permeability simulations were run in a MRV of 300^3^ voxels^3^ that represents a 7.4 mm^3^ of the sample. The MRV was virtually cropped from the center of the sample to avoid the border effects.

### 2.4. Compression Tests

The mechanical strength of the Ti6420Ag porous composites was evaluated by simple compression tests. For that, samples with a 6 mm diameter and 8 mm height were sintered. The tests were carried out with a 1150 Instron Universal machine, following the ASTM D695-02 requirements. Tests were performed with a strain rate of 0.5 mm·min^−1^, as reported elsewhere [37]. The data collected by the software of the Instron during the tests were the displacement and load. These data obtained from the compression tests were used to calculate the strain and the compression stress, using the initial dimensions of the samples. The Young’s modulus (E) was obtained by fitting a straight line in the elastic zone of the curves. The Yield stress was also determined from the stress–strain curves by taking into account the values used for calculating the E values.

## 3. Results and Discussion

### 3.1. Dilatometry Analysis

The sintering behavior of the Ti64/20Ag with and without pore formers is shown in Figure 1. The axial strain of the samples is very similar to a thermal expansion up to 750 °C, where a small swelling is observed (Figure 1a). The maximum dilation was obtained at 860 °C, and then a sharp shrinkage was observed in all the samples until the cooling stage was reached. Finally, a linear behavior was observed during the cooling down period to room temperature. From the strain rate as a function of temperature, it is possible to notice that the axial deformation is mainly due to the liquid Ag formation. There are two sharp increments in the strain rate, at 940 °C and 962 °C, that accomplish most of the densification of the samples. At 940 °C, the softening of Ag particles can generate a densification by rearrangement since the melting point is close. At 962 °C, the liquid Ag can fully spread among the interparticle pores left by the Ti64 particles, as it was reported in [34]. The strain rate could also confirm that the addition of large pores does not change the behavior of the densification by liquid phase. At the end, the shrinkage (negative axial strain) increased along with the pore former volume. This could suggest that the liquid Ag drains into the large pores; however, this is not true, as is shown in Figure 2. Therefore, the additional shrinkage obtained from the samples is due to the pore deformation during sintering. This is possible because the densification generates compression stresses that can reduce the pore size initially created by the pore formers.

The densities of all samples are listed in Table 1, and it can be noticed that a reduction in the values is obtained. The sample with 50 vol.% of pore formers shows a sintered density of 3.2 g/cm^3^, which is a lower value of the Ti64 alloy and close to the ideal value reported for metallic implants elsewhere of 1.8 g/cm^3^ [38]. The densification of the porous samples is larger than the one without pore formers, which increases as the pore former’s volume does, too. However, the difference in densification is lower in comparison to the one measured by the axial strain. This suggests that deformation of large pores during sintering is anisotropic, which can be associated with the pore shape and orientation, as was pointed out by Cabezas et al. [8].

### 3.2. Microstructure Analysis

The microstructure observed by SEM is shown in Figure 2. Three different characteristics can be observed from Figure 2a: the large pores, the Ti64 matrix in dark grey and the Ag in light grey. The large pores are elongated and randomly distributed in the sample. The distribution of Ag around the Ti64 particles is better observed in Figure 2b. Figure 2c shows the microstructure inside of a large pore, in which it is possible to confirm that the Ag filled the interparticle pores but not the large pores. Figure 2d shows the surface of the particles exposed at the surface of large pores, in which the Ag is confined by capillarity between the smaller pores. It can also be noticed that the surface of the particles is not covered by Ag, which confirms that the liquid only densified the porosity left by the particle packing. In addition, it was confirmed that some Ag diffuses in a solid state into the Ti64 net to form thin lamellae of the Ti_2_Ag intermetallic (Figure 2e,f), as was reported for alloys processed by casting [20,22] or composites fabricated by powder metallurgy [34].

In order to observe the distribution of the Ag in the porous composite and quantify the quantity of the volume occupied by the liquid, 3D images with a 0.7 µm pixel resolution were acquired (Figure 3). Figure 3a shows a 2D virtual slice of the sample with 30 vol.% of pore formers, and three different gray levels can be distinguished in the image. Very dark grey is associated with the pores, the dark grey corresponds to the Ti64 and the light grey corresponds to the Ag. In order to segment the different phases, a thresholding procedure is performed by selecting the gray level value that corresponds to each phase in the image, as indicated in Figure 3e. Thanks to the good contrast generated by the difference in densities of the Ti64 and Ag, it is possible to obtain binary images that represent each phase, Ti64—Figure 3b, Ag—Figure 3c and pores—Figure 3d.

A 3D rendering of each phase is shown in Figure 4, which allows us to qualitatively determine the interaction between Ti64, Ag and pores in 3D. However, the quantitative data are obtained from binary images, for example, the volume fraction of each phase is determined by counting the voxels corresponding to the Ti64, Ag and pores, respectively, and dividing each one by the total volume. However, these kinds of images were acquired mainly to quantitatively determine the distribution of Ag in the composite and not the pores because only a small volume is observed. From these images, the average volume fraction of Ag measured from 3D images of samples with different quantities of pore formers was 22%, which is close to the 20 vol.% added. The small difference can be attributed to the image treatment that could slightly overestimate the Ag. The interconnectivity of the Ti64 and Ag was determined by following the procedure described elsewhere [8]. It was determined that Ti64 is fully interconnected in 3D, and the Ag shows a 96% interconnectivity with the small, isolated volumes. From the 3D image analysis, it was also demonstrated that liquid Ag remains into the interparticle pores formed by the Ti64 packing and that liquid does not drain into the large pores.

### 3.3. Porosity Analysis and Permeability

The 2D virtual slices of Ti64/20Ag samples with different volume fractions of pore formers acquired with a pixel resolution of 6.5 µm allow us to observe the homogeneous distribution of large pores in the whole sample (Figure 5). It can also qualitatively be observed that no pore agglomeration or segregation is found, even in the sample containing 50 vol. % of pore formers. In order to obtain the quantitative data, the grey level images were transformed into binary ones following a similar process as described above, with the difference being that the Ag cannot be detected with this pixel size resolution. In this case, only the pores and the solid that includes Ti64 and Ag were segmented, but the main purpose of such images was to evaluate the porosity. The volume fraction of pores was determined as mentioned above, and the values are listed in Table 2. For all the samples, a higher value of porosity than the volume of pore formers added is found. The difference between the added value and the found one might be due to the packing arrangement during powder compaction. This is because the coordination number of the salt particles increases, which leads to larger interparticle pores among the salt particles that could not be filled by the Ti64 and Ag particles.

The pore connectivity is determined by analyzing the continuity of pores inside the measured volume; this procedure is detailed in [39]. The values obtained were higher than 95% for all the samples, which can also be qualitatively observed in Figure 6b,e,h. A 3D rendering of the porosity in the analyzed volumes is presented, in which pseudo-colors indicate the interconnected pores. Thus, each color in the image indicates a group of pores interconnected among them. It can be noticed that one color is predominant in the image, but also that more isolated pores are found in the sample with 30 vol.% of pore formers (Figure 6b).

The pore size distribution of the samples with different quantities of pore formers is very similar, which confirms a good distribution of the pore formers as well as the repeatability of the process (see Figure 7). The pore sizes go from 30 to 200 µm for every sample. Moreover, the median pore size (d_50_) presents similar values ranging from 75 to 89 µm (see Table 2), that suggests a good distribution of the pore formers during the whole fabrication process. Furthermore, the median pore size also indicates a reduction of 47% with respect to the median size of the initial pore sizes (140 µm). This can support the discussion about the pore deformation undergone by the samples during sintering, as mentioned above. The 3D rendering of the samples with different quantities of pore volumes illustrates the increment in the void spaces but with similar size (Figure 6a,d,g). The pore size distribution finds itself in the range reported as optimal to favor the cell ingrowth by Itälä et al. (100–200 µm) [40]. However, the median pore size we obtained is low in comparison to the one used by Pérez et al. to regenerate the bone defects, with macropores of 210 µm on average [12].

Figure 6c,f,i shows the streamlines of simulated fluid passing throughout the pores, which are color-coded according to the relative fluid velocity. The calculated permeability values are listed in Table 2 for the Ti64/20Ag samples with different quantities of pore formers. An increment of eight times was found when the pore volume fraction of samples increased from 33 to 57% and for a small difference in the pore size. This can be associated to the reduction in the tortuosity from a value of 1.83 to 1.32 (see Table 2). However, this increment in the permeability was also obtained because an increment in the porosity generates more paths for the fluid to pass throughout the porous sample, as can be qualitatively observed from Figure 6c,f,i, in which there are more paths for the sample with 50 vol.% of pore formers, which are also low-tortuous, allowing higher velocities of the fluid, although the pore connectivity is the same with respect to the sample with 40 vol.% of pore formers. The permeability values of the samples are between 0.47 and 3.93 × 10^−11^ m^2^, which are in the range of the human trabecular bones in the proximal femur (3 × 10^−11^ to 5 × 10^−10^ m^2^), but low compared to the values for the permeability of the human vertebral trabecular bone, 10^−8^ to 10^−9^ m^2^ [13]. Permeability can be increased by the addition of higher quantities of pore formers, but this can compromise the structure of the Ti64/20Ag, as the liquid Ag may collapse the thinner struts obtained. The second method to improve permeability and obtain values closer to the human trabecular bones is to increase the pore former’s size. Additional tests should be carried out to demonstrate whether one of these two methods could be viable to increase the permeability. Nevertheless, the obtained values of permeability are in the range of different scaffolds fabricated by space holder [41] or AM [42] with similar volume fraction of pores.

### 3.4. Compression Analysis

The behavior under simple compression of the Ti64/20Ag samples with and without pore formers was plotted in Figure 8. It was found that the sample without pore formers shows a higher strength in comparison to the porous samples. Further, all the samples show an elastic zone, then a plastic deformation with a small increment in the stress and, then, a plastic deformation at more or less constant stress until the failure. As was expected, the addition of large pores generates a high reduction in the maximal compression strength of samples from 916 MPa to a maximum of 117 MPa.

The Young’s modulus (E) was obtained from the elastic zone of the curves, and it is listed in Table 3, as well as the yield stress (σ_y_). It was found that E is reduced seven times by adding 50 vol.% of pore formers, meanwhile, the σ_y_ is also reduced, but only five times. The E values of the Ti64/20Ag porous samples are in the range of the human bones, under 20 GPa [4]. The σ_y_ values are also in the range reported for human bones (50–200 MPa), although the sample with 30 vol.% of pore formers is a little higher, 220.9 MPa. Thus, the compression values are close to the ones reported for Ti64 porous materials fabricated by space holder [43] or AM [44] with similar volume fraction of pores, however, there are no reports of Ti64Ag porous materials to directly compare with our results. It was also pointed out by Liu et al. [45] that the σ_y_/E ratio, called “admissible strain”, is a good parameter to improve the performance of the metallic implants, and it is recommended that this value should be the highest possible.

The σ_y_/E was calculated and listed in Table 3 for all the samples. The values of σ_y_/E range from 12 to 16 × 10^−3^ and the highest value corresponds to the Ti64/20Ag sample with 50 vol.% of pore formers. In conclusion, these values are higher than the ones reported for the solid Ti64 [46] but lower than the ones obtained for porous composites of Ti64/xTa [47]. More, the σ_y_/E values are in the range of the compact bones (11 × 10^−3^) but lower than the ones reported for the human vertebral trabecular bone (35 × 10^−3^) [3].

## 4. Conclusions

An innovative strategy to fabricate Ti64/20Ag highly porous composites by the space holder technique and consolidated by liquid phase sintering, at lower temperatures than the ones used for Ti64 materials, was successfully studied. It was demonstrated that the liquid Ag spreads over the interparticle pores, densifying the composite without filling the large pores. Full pore connectivity was obtained, with a pore size distribution controlled by the initial size of the pore formers and with the help of the sintering densification. The porous samples presented the characteristics of the pores, the permeability and the mechanical properties that make them suitable for bone implants. However, the best results for such applications were obtained for the Ti64/20Ag sample with 50 vol.% of pore formers, as its structural properties come closest to those of human bones. Additional corrosion, antibacterial and biological tests are in progress to establish the benefits of the Ag addition in Ti64 porous materials.

## Figures and Tables

**Figure 1 materials-15-05956-f001:**
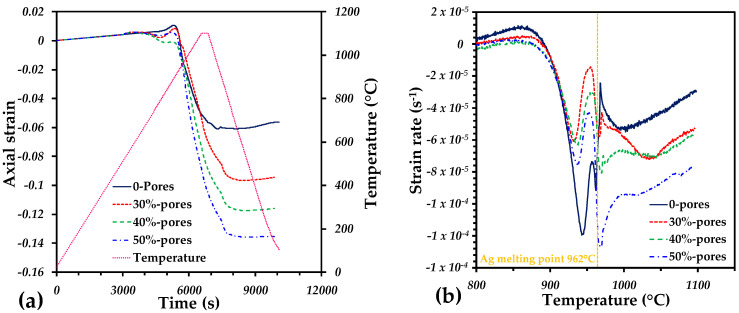
Dilatometry curves: (**a**) evolution of the axial strain and temperature as a function of time during sintering; (**b**) strain rate as a function of the temperature during the densification of samples.

**Figure 2 materials-15-05956-f002:**
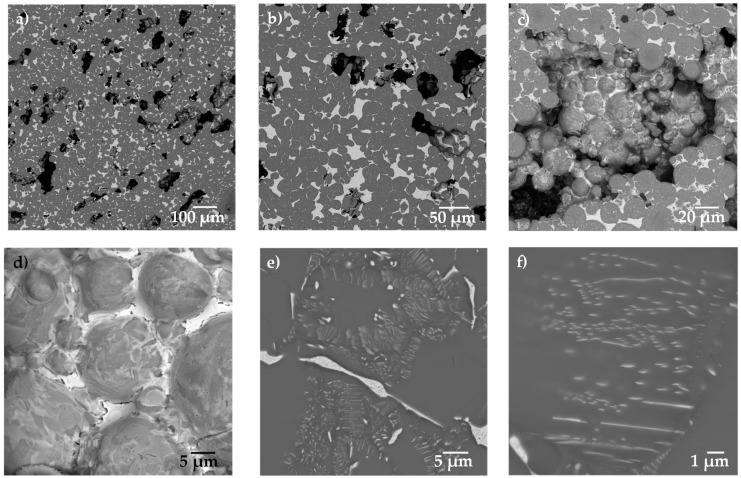
Backscattered electron image of the Ti64/20Ag sample with 30 vol.% of pore formers at different magnifications: (**a**) 200×, (**b**) 500×, (**c**) 1000×, (**d**) 2000×, (**e**) 5000× and (**f**) 10,000×.

**Figure 3 materials-15-05956-f003:**

2D virtual slices of the Ti64/20Ag sample with 30 vol.% of pore formers after sintering: (**a**) grey level image, (**b**–**d**) binary images of Ti64, Ag and pores, respectively. (**e**) Histogram of voxel gray levels of the complete 3D image.

**Figure 4 materials-15-05956-f004:**
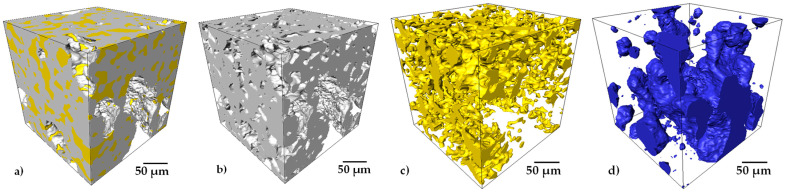
A 3D rendering of a virtually cropped volume showing the interaction of the three phases (**a**) and tridimensional distributions of the individual phases, (**b**) Ti64, (**c**) Ag and (**d**) pores.

**Figure 5 materials-15-05956-f005:**
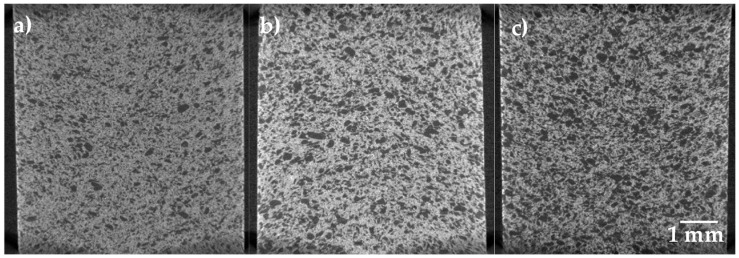
2D virtual slices of the Ti64/20Ag samples after sintering at 1100 °C, acquired with a pixel resolution of 6.5 µm and with different quantities of pore formers: (**a**) 30 vol.%, (**b**) 40 vol.% and (**c**) 50 vol.%.

**Figure 6 materials-15-05956-f006:**
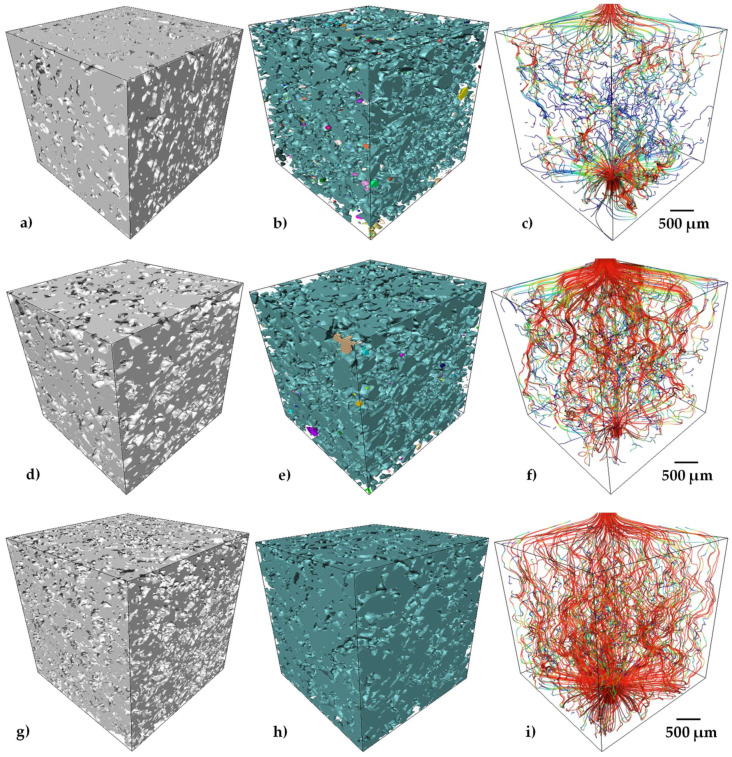
3D volume renderings of solid Ti64/20Ag, porosity and the streamlines for the simulated flow of fluid through the void space in samples. The colors of the streamlines illustrate the fluid velocity for Ti64/20Ag samples with 30 vol.% (**a**–**c**), 40 vol.% (**d**–**f**) and 50 vol.% (**g**–**i**) of pore formers.

**Figure 7 materials-15-05956-f007:**
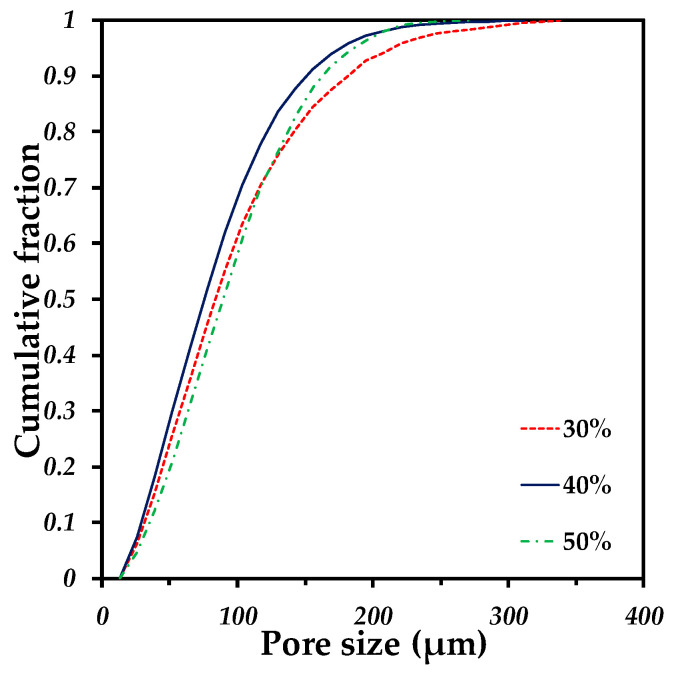
Pore size distribution of Ti64/20Ag samples containing different volume fractions of pore formers.

**Figure 8 materials-15-05956-f008:**
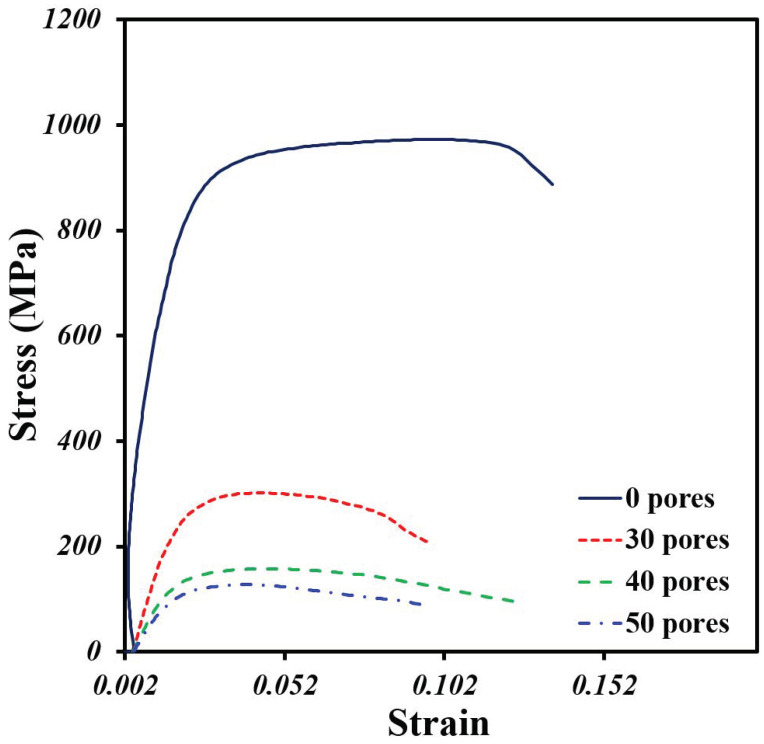
Stress–strain compression curves of Ti64/20Ag samples with different volume fractions of pore formers.

**Table 1 materials-15-05956-t001:** Green and sintered densities of all samples as well as densification reached after sintering.

Vol.% of Salts	ρ_0_ (g/cm^3^)	ρ_s_ (g/cm^3^)	(ρ_s_−ρ_0_)/ρ_0_
0	4.57	5.42	0.18
30	3.39	4.27	0.25
40	3.16	4.07	0.28
50	2.48	3.20	0.29

**Table 2 materials-15-05956-t002:** Pore characteristics of Ti64/20Ag samples fabricated with different vol.% of pore formers.

Volume Fraction of Pore Formers (%)	Volume Fraction of Pores (%)	Pore Connectivity (%)	Median Pore Size (µm)	Permeability (10^−11^ m^2^)	Tortuosity
30	33.38	95.58	83.79	0.47	1.83
40	45.04	99.12	75.43	1.33	1.51
50	57.49	99.85	89.11	3.93	1.32

**Table 3 materials-15-05956-t003:** Mechanical properties of Ti64 and Ti64/20Ag composites.

Vol.% of Salts	E (Gpa)	σ_y_ (Mpa)	σ_y_/E (10^−3^)
0	49.4	597.9	12.1
30	18.1	220.9	12.2
40	10.6	154.6	14.5
50	7.4	123.6	16.7

## Data Availability

The raw/processed data required to reproduce these findings cannot be shared at this time as the data also form part of an ongoing study.

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
