# Peer review of "Ti64/20Ag Porous Composites Fabricated by Powder Metallurgy for Biomedical Applications"

_materials, 2022, doi:10.3390/ma15175956_

Round 1

Reviewer 1 Report

This paper presented a highly porous composite Ti64/20Ag fabricated by powder metallurgy for biomedical applications, and mainly investigated its microstructure and mechanical proprieties. The research contents are sufficient, and can be considered for publication in materials. However, the following corrections should be also considered by authors before publication in the journal.

(1) In the manuscript, the X ray diffraction analysis should be introduced to analyzed the phase constitutions of the porous materials.

(2) In the introduction section, the authors claimed that the optimal concentration of Ag in Ti is between 20 and 25 wt.% and Ag in Ti64 alloy is 20 wt%, but their elastic modulus are much bigger, which are disadvantaged for the application as bone implants. The authors should give the corresponding elastic modulus, and compare with the experimental results in this manuscript.

(3) Generally speaking, A systematic experiments should include five experiment samples. However, this manuscript selected only three experiment samples (30 vol.%, 40 vol.% and 50 vol.%.), 35 vol.% and 45 vol.% should be considered.

(4) Figure 5 shows the 2D virtual slices of the Ti64/20Ag samples after sintering at 1100°C, a partial enlargement of the figures should be provided.

(5) In table.3, the mechanical properties of Ti64 and Ti64/xCu composites, the second line 0 vol.% of salts, the Young’s modulus E (Gpa) should be 49.4.

(6) Line 359 and line 360, the σy/E values (11 10-3) should be corrected as (11×10-3), and the other number should be corrected as well.

(7) The text contains some language errors and some obvious mistakes. A careful check is needed.

Author Response

This paper presented a highly porous composite Ti64/20Ag fabricated by powder metallurgy for biomedical applications, and mainly investigated its microstructure and mechanical proprieties. The research contents are sufficient, and can be considered for publication in materials. However, the following corrections should be also considered by authors before publication in the journal.

(1) In the manuscript, the X ray diffraction analysis should be introduced to analyzed the phase constitutions of the porous materials.

The X ray diffraction analysis of the Ag addition was described in a previous paper, thus, in this one we are referring to such analysis:

34. Solorio, V.M., Vergara-Hernández, H.J., Olmos, L., Bouvard, D., Chávez, J., Jimenez, O., Camacho, N. Effect of the Ag addition on the compressibility, sintering and properties of Ti6Al4V/xAg composites processed by powder metallurgy. Journal of Alloys and Compo J. Alloys Compd., 2022, 890, 161813.

(2) In the introduction section, the authors claimed that the optimal concentration of Ag in Ti is between 20 and 25 wt.% and Ag in Ti64 alloy is 20 wt%, but their elastic modulus are much bigger, which are disadvantaged for the application as bone implants. The authors should give the corresponding elastic modulus, and compare with the experimental results in this manuscript.

We modified the manuscript, the corresponding E values are now indicated within the paper.

(3) Generally speaking, A systematic experiments should include five experiment samples. However, this manuscript selected only three experiment samples (30 vol.%, 40 vol.% and 50 vol.%.), 35 vol.% and 45 vol.% should be considered.

It is possible to have 5 different conditions, however, a difference of 5 vol.% of pore formers doesn’t have enough effect to be considered. Instead, 60 and 70 vol.% should be more representative, nevertheless, by this technique to obtain such highly porous materials is experimentally complicated.

(4) Figure 5 shows the 2D virtual slices of the Ti64/20Ag samples after sintering at 1100°C, a partial enlargement of the figures should be provided.

This tomography images cannot be enlarged since the quality is highly reduced, an enlargement of such images is showed in Fig. 3. But it is necessary to acquire images with a low pixel size.

(5) In table.3, the mechanical properties of Ti64 and Ti64/xCu composites, the second line 0 vol.% of salts, the Young’s modulus E (Gpa) should be 49.4.

Thank you for the commentary, this was modified.

(6) Line 359 and line 360, the σy/E values (11 10-3) should be corrected as (11×10-3), and the other number should be corrected as well.

The units were modified as suggested.

(7) The text contains some language errors and some obvious mistakes. A careful check is needed.

The whole manuscript was carefully checked and corrected.

Reviewer 2 Report

The authors investigated the microstructure and mechanical properties of Ti64/20Ag porous composite fabricated by space holder technique and liquid state sintering for biomedical applications. The manuscript was an interesting topic and well written, but It could be accepted after minor revision as below:

11. For the abstract, it is suggested to state the summary value for the result of the study. For example, the value of porosity and compression.

  2. Refer to line 50, “fabrication of composites by AM is more complicated than by conventional sintering”. Could you go into further detail about the limitations of AM against traditional sintering? What is AM method you mean?

33.  Why select Ag over other materials?

44.  Line 52, “Ti64 highly porous have been obtained by space holder reaching stiffness values similar to the human bones” was missing the references.

55. Refer to line 154,155: “The numerical simulations are based on the Darcy law, in which the simulation considers an incompressible Newtonian fluid with a steady state laminar flow and a viscosity of 0.045 Pa-s, representing the viscosity of the blood”, Did you cite any previous studies on this?

66. For Figure 7, please use standard English for x and y axis title.

77. Line 354, “Table 3 Mechanical properties of Ti64 and Ti64/xCu composites”. Which data are Ti64 and Ti64/xCu? Why Ti64/xCu? Is that Ag or Cu?

88. For the result of porosity and compression, it is highly suggested to do ANOVA analysis. It is useful to identify the significant factor contributing to the experimental conditions. You can refer to this paper (Application of Taguchi Method to Optimize the Parameter of Fused Deposition Modeling (FDM) Using Oil Palm Fiber Reinforced Thermoplastic Composites), which can assist you in performing ANOVA analysis. 

Author Response

The authors investigated the microstructure and mechanical properties of Ti64/20Ag porous composite fabricated by space holder technique and liquid state sintering for biomedical applications. The manuscript was an interesting topic and well written, but It could be accepted after minor revision as below:

  1. For the abstract, it is suggested to state the summary value for the result of the study. For example, the value of porosity and compression.

The abstract was modified as suggested.

  1. Refer to line 50, “fabrication of composites by AM is more complicated than by conventional sintering”. Could you go into further detail about the limitations of AM against traditional sintering? What is AM method you mean?

We are comparing with the classic AM Laser methods.

  1. Why select Ag over other materials?

Because the Ag is immiscible with titanium, allowing to reduce the sintering temperature without modifying the solid structure.

  1. Line 52, “Ti64 highly porous have been obtained by space holder reaching stiffness values similar to the human bones” was missing the references.

Thank you very much for your observation. References were added.

  1. Refer to line 154,155: “The numerical simulations are based on the Darcy law, in which the simulation considers an incompressible Newtonian fluid with a steady state laminar flow and a viscosity of 0.045 Pa-s, representing the viscosity of the blood”, Did you cite any previous studies on this?

This is how the Avizo software works in the numerical simulations.

  1. For Figure 7, please use standard English for x and y axis title.

Thank you. This was corrected.

  1. Line 354, “Table 3 Mechanical properties of Ti64 and Ti64/xCu composites”. Which data are Ti64 and Ti64/xCu? Why Ti64/xCu? Is that Ag or Cu?

This was a mistake, and it was corrected.

  1. For the result of porosity and compression, it is highly suggested to do ANOVA analysis. It is useful to identify the significant factor contributing to the experimental conditions. You can refer to this paper (Application of Taguchi Method to Optimize the Parameter of Fused Deposition Modeling (FDM) Using Oil Palm Fiber Reinforced Thermoplastic Composites), which can assist you in performing ANOVA analysis.

Thank you very much for the suggestion, this can be a helpful tool. However, we believe that for this paper is not possible to do this analysis from the obtained data and on a short notice, but we will consider it for future works.

Round 2
